# Improved Performance for PMSM Sensorless Control Based on Robust-Type Controller, ESO-Type Observer, Multiple Neural Networks, and RL-TD3 Agent [note 1]

**DOI:** 10.3390/s23135799

**Published:** 2023-06-21

**Authors:** Marcel Nicola, Claudiu-Ionel Nicola, Cosmin Ionete, Dorin Șendrescu, Monica Roman

**Affiliations:** 1Research and Development Department, National Institute for Research, Development and Testing in Electrical Engineering—ICMET Craiova, 200746 Craiova, Romania; 2Department of Automatic Control and Electronics, University of Craiova, 200585 Craiova, Romaniacosmin.ionete@edu.ucv.ro (C.I.); dorin.sendrescu@edu.ucv.ro (D.Ș.); monica.roman@edu.ucv.ro (M.R.)

**Keywords:** PMSM, robust control, extended state observer, neural networks, reinforcement learning

## Abstract

This paper summarizes a robust controller based on the fact that, in the operation of a permanent magnet synchronous motor (PMSM), a number of disturbance factors naturally occur, among which both changes in internal parameters (e.g., stator resistance *R_s_* and combined inertia of rotor and load *J*) and changes in load torque *T_L_* can be mentioned. In this way, the performance of the control system can be maintained over a relatively wide range of variation in the types of parameters mentioned above. It also presents the synthesis of robust control, the implementation in MATLAB/Simulink, and an improved version using a reinforcement learning twin-delayed deep deterministic policy gradient (RL-TD3) agent, working in tandem with the robust controller to achieve superior performance of the PMSM sensored control system. The comparison of the proposed control systems, in the case of sensored control versus the classical field oriented control (FOC) structure, based on classical PI-type controllers, is made both in terms of the usual response time and error speed ripple, but also in terms of the fractal dimension (DF) of the rotor speed signal, by verifying the hypothesis that the use of a more efficient control system results in a higher DF of the controlled variable. Starting from a basic structure of an ESO-type observer which, by its structure, allows the estimation of both the PMSM rotor speed and a term incorporating the disturbances on the system (from which, in this case, an estimate of the PMSM load torque can be extracted), four variants of observers are proposed, obtained by combining the use of a multiple neural network (NN) load torque observer and an RL-TD3 agent. The numerical simulations performed in MATLAB/Simulink validate the superior performance obtained by using properly trained RL-TD3 agents, both in the case of sensored and sensorless control.

## 1. Introduction

In today’s world, the control of actuation systems based on servomotors has become a priority, due to the fact that these systems can be found in a wide range of products that contribute both to improving the efficiency of certain activities and to improving ergonomics [1,2]. It is worth remembering that PMSM is incorporated into servomotors used in robotics, computer peripherals, the aerospace industry, electric drives, etc. [3,4]. Thus, PMSM control takes on new dimensions, in terms of the complex concerns of researchers to improve the performance of control systems. It is worth remembering that the FOC control strategy is essential for PMSM control, and ensures superior control performance by organizing the control system into two cascaded levels, an inner level of current control and an outer level of PMSM rotor control [5,6]. Typically, the controllers used within the FOC control strategy structure are PI controllers [7,8]. The relatively good performance of these types of controllers is well-known, but when significant parametric variations occur, the performance of these types of control systems, based on PI-type controllers, decreases significantly. A number of PMSM control systems have been developed and adapted, including adaptive [9,10], optimal [11,12] and predictive [13,14], but also fuzzy [15,16] and neuro-fuzzy control systems have been invented with good results [17,18]. A backstepping controller for IM is developed in [19].

The sensorless control is also used in modern PMSM control systems to increase the reliability and operational safety of the PMSM, by using PMSM rotor speed observers and eliminating the speed sensor. Common observers include the Luenberger observer [20], the sliding mode observer (SMO) [21], the model reference adaptive system (MRAS) [22] for the deterministic case, and the Kalman filter [23] for the stochastic case.

Since there are significant variations in both the parameters describing the operation of the PMSM and the load torque during normal operation of the PMSM, robust control systems [24,25,26,27,28] may be an ideal choice of control system for the PMSM.

This paper is a continuation of the paper presented in [28] and focuses on extending the performance of the sensored control system using a robust controller type, and also in the case of the sensorless control system using an ESO-type observer [29]. This type of observer is designed by exploiting the form of the differential equations describing the PMSM. Thus, it is possible to obtain both PMSM rotor speed estimation and load torque estimation, since the ESO-type observer incorporates the disturbances acting on the system (e.g., load torque) in an additional state variable. This additional state variable is also estimated by the ESO-type observer. In terms of methods to improve the PMSM rotor speed estimation using the ESO-type observer structure, it is proposed to use this observer in combination with a multiple NN [30,31] load torque observer and an RL-TD3 agent [32,33] to obtain a better performance on rotor speed estimation compared to the measured rotor speed provided by a speed sensor.

The paper presents the main control structures and observers of the PMSM, starting with the differential equation description, the robust controller synthesis, and the ESO-type observer synthesis that performs the PMSM rotor speed estimation. Improved variants of using the ESO-type observer in tandem with the multiple NN load torque observer and RL-TD3 agent are also presented. In addition, the control structures, the reward and the training stage for the multiple NN load torque observer and RL-TD3 agent for speed estimation are presented.

The main contributions of this article are:The synthesis of a robust PMSM controller using the *d-q* frame mathematical model for the sensored case of the PMSM control structure;The robust controller synthesis using MATLAB and integration with PMSM sensored control system;The improvement in the control performance of the PMSM sensored control system, by combining the robust controller and an RL-TD3 agent that provides additional correction signals to adjust the *u_d_* and *u_q_* commands generated by the main robust controller;The comparative presentation of the performance of the proposed controllers used in the PMSM control system structure, in terms of response time, speed ripple error and DF of the rotor speed signal;The ESO-type observer synthesis for the sensorless case of the PMSM control system for rotor speed estimation;The improvement in the performance of the basic ESO observer structure for PMSM rotor speed estimation, by providing load torque estimates using two-layer feed-forward multiple NN networks implemented with the Neural Net Fitting MATLAB application [31];The improvement in the performance of the ESO basic observer structure for PMSM rotor speed estimation, by implementing an RL-TD3 agent [33] in MATLAB/Simulink that provides correction signals to estimate the PMSM rotor speed value as close as possible to the sensored rotor speed provided by a PMSM speed sensor;Comparison of the performance of the four observer variants for estimating the proposed PMSM rotor speed, in terms of response time and speed ripple error.

The rest of the article is organized as follows: Section 2 presents the proposed sensored and sensorless control structure of PMSM. The robust control of the PMSM is presented in Section 3, while the MATLAB/Simulink implementation of the PMSM sensored control system together with the improvement in the control performance by using an RL-TD3 agent is presented in Section 4. The PMSM sensorless control system using improved ESO-type observer variants is presented in Section 5, and numerical simulations of the operation of the PMSM sensorless control system are presented in Section 6. Section 7 presents some conclusions and suggestions for future approaches.

## 2. Sensored and Sensorless PMSM Control

Traditionally, PMSM control is achieved using the FOC strategy, in which a PMSM rotor speed sensor is integrated and the name of the control structure is sensored control. Figure 1 shows the schematic diagram of the FOC-type strategy for the PMSM control system, where the controllers used in the current and voltage loops are PI-type.
(1)$$\left\{\begin{array}{l} \frac{di_{d}}{dt}=-\frac{R_{s}}{L_{d}}i_{d}+\frac{L_{q}}{L_{d}}n_{p}\omega i_{q}+\frac{1}{L_{d}}u_{d}\\ \frac{di_{q}}{dt}=-\frac{R_{s}}{L_{q}}i_{q}-\frac{L_{d}}{L_{q}}n_{p}\omega i_{d}-\frac{\lambda_{0}}{L_{q}}n_{p}\omega +\frac{1}{L_{q}}u_{q}\\ \frac{d\omega }{dt}=\frac{3}{2}\frac{n_{p}}{J}\left(\lambda_{0}i_{q}+\left(L_{d}-L_{q}\right)\;i_{d}i_{q}\right)-\frac{1}{J}T_{L}-\frac{B}{J}\omega \\ \frac{d\theta_{e}}{dt}=n_{p}\omega \end{array}\right.$$

In Relation (1), the notations are the usual ones in the *d*-*q* rotating reference frame [1,2,28]. Therefore, it can be denoted these notations as follows: the stator voltages, *u_d_* and *u_q_*, in the *d*-*q* frame; the stator currents, *i_d_* and *i_q_*, in the *d*-*q* frame; the stator inductances, *L_d_* and *L_q_*, in the *d*-*q* frame; and the stator resistances, *R_d_* and *R_q_*, in the *d*-*q* frame. Other common parameters of the PMSM include: the flux linkage is denoted by *λ*_0_, the number of pole pairs is denoted by *n_p_*, the rotor moment inertia combined with load moment inertia is denoted by *J*, the load torque is denoted by *T_L_*, the viscous friction coefficient is denoted by *B*, and the PMSM rotor speed is noted with *ω*.

In short, the description of the nonlinear operating equations of the PMSM is given by Relation (1).

In the usual case, where *L_d_* = *L_q_* is *R_d_* = *R_q_* = *R_s_*, the system shown in (1) can be rewritten in the following form [1,2,28]:(2)$$\left[\begin{array}{c} {\dot{i}}_{d}\\ {\dot{i}}_{q}\\ \dot{\omega } \end{array}\right]=\left[\begin{array}{c} -\frac{R_{s}}{L_{q}}i_{d}+n_{p}\omega \;i_{q}\\ -\frac{R_{s}}{L_{q}}i_{q}-n_{p}\omega \;i_{d}-\frac{\lambda_{0}}{L_{q}}n_{p}\omega \\ \frac{3}{2}\frac{n_{p}\lambda_{0}}{J}i_{q}-\frac{B}{J}\omega \end{array}\right]+\left[\begin{array}{cc} \frac{1}{L_{q}} & 0\\ 0 & \frac{1}{L_{q}}\\ 0 & 0 \end{array}\right]\left[\begin{array}{c} u_{d}\\ u_{q} \end{array}\right]+\left[\begin{array}{c} 0\\ 0\\ -\frac{T_{L}}{J} \end{array}\right]$$

In the following sections we propose both an improved control structure, in the case of sensored control, and the use of ESO-type observer combinations, in the case of a structure called sensorless control. Thus, Figure 2 shows the schematic diagram of the proposed PMSM sensorless control system based on a robust controller and RL-TD3 agent algorithm, using an improved ESO-type observer combined with a multiple NN load torque estimation and an RL-TD3 agent for speed estimation.

In terms of improving control structures, a robust controller combined with an RL-TD3 agent is presented, which provides superior performance compared to classical PI controllers.

Four combinations of observers are proposed for PMSM speed estimation. The first type of observer is based on the ESO-type observer, which can estimate not only the PMSM rotor speed *ω*, but also the load torque *T_L_*, by adding another variable containing disturbance elements and by a special structure of the observer. A second proposed variant is a multiple NN, trained over different load torque ranges, which provides this value to an ESO-type observer which, in turn, will have a better PMSM rotor speed estimation relative to the measured PMSM rotor speed (sensored control structure case). The following two variants are derived from those presented above, with the addition of an RL-TD3 agent that provides superior estimates of rotor PMSM.

## 3. Robust Control of PMSM

Consider the operating point (*x**, *u**) obtained from the linearization of the nonlinear PMSM model in Section 2. Consider the operating point (*x**, *u**) obtained by linearizing the nonlinear PMSM model in Section 2. The explanation of the operating point is given by the following Relation (3) [28].
(3)$${\underline{x}}^{*}={\left[\begin{array}{ccc} i_{d}^{*} & i_{q}^{*} & \omega^{*} \end{array}\right]}^{T}\;;\;{\underline{u}}^{*}={\left[\begin{array}{cc} u_{d}^{*} & u_{q}^{*}\;\left|\;T_{L}\right. \end{array}\right]}^{T}$$

Around the operating point (*x**, *u***)*, the nonlinear equation of the PMSM takes the following form:(4)$$f\left(x^{*},u^{*}\right)=0$$

The following linearized system can be related to the nonlinear Equation (4):(5)$$\left\{\begin{array}{c} \dot{x}(t)=Ax(t)+Bw(t)+Du(t)\\ y(t)=Cx(t) \end{array}\right.$$
where the matrices are defined by Relation (6).
(6)$$A=\left[\begin{array}{ccc} -\frac{R_{s}}{L_{q}} & \omega^{*} & i_{q}^{*}\\ \omega^{*} & -\frac{R_{s}}{L_{q}} & \frac{-n_{p}\lambda_{0}}{L_{q}}\\ 0 & \frac{3}{2}\frac{n_{p}\lambda_{0}}{J} & -\frac{B}{J} \end{array}\right];\;B=\left[\begin{array}{ccc} \frac{1}{L_{q}} & 0 & 0\\ 0 & \frac{1}{L_{q}} & 0\\ 0 & 0 & -\frac{1}{J} \end{array}\right];C=\left[\begin{array}{ccc} 1 & 0 & 0\\ 0 & 0 & 1 \end{array}\right];\;D=\left[\begin{array}{ccc} 0 & 0 & 0\\ 0 & 0 & 0 \end{array}\right]$$

Relation (7) shows the reference values selected for the operating point:(7)$$\left[\begin{array}{c} i_{d}^{*}\\ i_{q}^{*}\\ \omega^{*} \end{array}\right]=\left[\begin{array}{c} 0\\ 1\\ 1200 \end{array}\right]$$

For robust control of the PMSM, Figure 3 schematically shows the control structure of the PMSM, showing the reference $r=\left[\begin{array}{c} i_{dref}\\ \omega_{ref} \end{array}\right]$, the measured vector $y=\left[\begin{array}{c} i_{d}\\ \omega \end{array}\right]$, and the load torque *T_L_*, which acts as a disturbance variable on the PMSM control system. After the synthesis of the robust control *K*, based on the synthesis of robust controllers, when the PMSM equations are described in the *d*-*q* rotating reference frame, the values of the control vector $u=\left[\begin{array}{c} u_{d}\\ u_{q} \end{array}\right]$ are obtained.

Following the steps in the synthesis of a robust control algorithm [23], Figure 4 shows the generalized plant. In addition to inputs/outputs and references, it contains the *K* controller structure, the nominal plant *P*, the uncertainties block ∆ and the input and output uncertainties *w* and *z*.

The relation between these elements can be written as follows:(8)w=Δ⋅z
(9)$$u=\left[\begin{array}{c} u_{d}\\ u_{q} \end{array}\right]=K\left[\begin{array}{c} r\\ y \end{array}\right]\;$$
where ∆ represents a diagonal matrix defining the uncertainties and *u* is the controller feedback, i.e., the command given by the robust controller *K*.

The global transfer matrix from *r* and *T_L_* to *y*−*r* is given by Relation (10):(10)$$y-r=F_{u}\left(F_{l}\left(P(s),K(s)\right),\Delta \right)\left[\begin{array}{c} r\\ T_{L} \end{array}\right]$$
where: the generalized plant is denoted by *P*, the robust controller is denoted by *K*, and the upper/lower fractional linear transform is denoted by *F_u/l_* [24,28].

According to [23], the generalized plant *P* verifies the following relation:(11)$$\left[\begin{array}{c} z\\ y-r\\ r\\ y \end{array}\right]=P\left[\begin{array}{c} w\\ r\\ T_{L}\\ u \end{array}\right]$$

Figure 5 shows the block diagram of the extended system. It also shows the weighting function *W_i_* (with *i* = 1, 2, 3) and the linearized nominal transfer function of the PMSM given in (6) and denoted by *G_nom_*.

Typically, the sensitivity function is denoted by *S*, the complementary sensitivity function is denoted by *T*, and the additive robustness matrix is denoted by *R*. Their expressions are given in (12)–(14) respectively.
(12)$$S=\frac{1}{1+G_{nom}K}$$
(13)$$T=\frac{KG}{1+G_{nom}K}\;$$
(14)$$R=\frac{K}{1+G_{nom}K}\;$$

The main objectives of robust control are tracking performance and disturbance attenuation over the entire frequency range of the controlled process. These objectives can be summarized as shown in Relation (15). The weighting matrices in Relation (15) are usually chosen as follows: *W_1S_* for a low-pass-type filter, *W_2U_* for a high-pass-type filter, and the matrix *W_3T_* provides high-frequency attenuation for the complementary sensitivity function *T* [24,28].
(15)$${\left\|T_{r\to z}\right\|}_{\infty }=\left\|\begin{array}{c} W_{1S}S\\ W_{2U}KS\\ W_{3T}T \end{array}\right\|<1$$

## 4. MATLAB/Simulink Implementation and Numerical Simulations for PMSM Sensored Control System Using a Robust Controller Combined with RL-TD3 Agent

Based on what was described in the previous section on the synthesis of a robust controller for the sensored control of PMSM, this section presents the MATLAB/Simulink implementation of such a controller, while the second part of this section presents a way to improve the performance of the robust controller using an RL-TD3 agent.

### 4.1. Robust Controller Synthesis Based on MATLAB/Simulink

The nominal parameters of the PMSM used in the following numerical simulations are shown in Table 1. Figure 6 shows the diagram of the MATLAB/Simulink model block for the proposed PMSM sensored control system using a robust controller.

The implementation of the robust controller follows the steps described in Section 3, as well as a set of instructions specific to robust synthesis, used in the MATLAB Robust Control toolbox environment [25]. Therefore, to define a variation in some parameters, e.g., load torque *T_L_*, around nominal values, the command *ureal*() is used. Using the *musyn*() command, the numerical values for the implementation of the robust controller K(s) are obtained as shown in Relation (16).


(16)
$$K\left(s\right)\left[\underline{\begin{array}{c} \underline{A_{K}}\\ C_{K} \end{array}\left|\begin{array}{c} \underline{B_{K}}\\ D_{K} \end{array}\right.}\right]\left(s\right)=\left[\begin{array}{ccccccccc} -2279 & 40.38 & 154.7 & 62260 & -1965 & -62180 & 1960 & 75.58 & -4.052\\ -362.8 & -1126 & -4392 & 1969 & 62160 & -1967 & -0.00062 & 303.2 & -52.76\\ 1037 & 131.3 & -648.5 & 0.00006 & -0.01642 & -0.00006 & 0.01639 & -1037 & 638.2\\ -5.336 & -0.0000002 & -0.02882 & -0.713 & -0.000004 & 0.2849 & 0.000004 & -2.664 & 0.02882\\ -94.65 & -0.000007 & -0.7645 & -0.000004 & -0.7119 & 0.000004 & 0.2838 & 94.65 & -7.236\\ 5.362 & 0 & 0.02779 & 0 & 0 & -1 & 0 & 2.638 & -02779\\ 93.5 & 0 & 0.8716 & 0 & 0 & 0 & -1 & -93.5 & 7.128\\ -15.88 & -0.5068 & 1.272 & 529.2 & -16.7 & 528.6 & 16.66 & 0 & 0\\ -0.5068 & -6.669 & -37.08 & 16.74 & 528.4 & -16.72 & -527 & 0 & 0 \end{array}\right]\left(s\right)$$


Furthermore, the controller *K* synthesized in Relation (16) and partitioned as shown in Relation (17) is also implemented in Simulink, as shown in Figure 6 [26,28].
(17)$$K(s)=\left[\begin{array}{cc} K_{11}(s) & K_{12}(s)\\ K_{21}(s) & K_{22}(s) \end{array}\right]$$
where each component of the controller *K*(*s*) can be written as a transfer function: num(s)/den(s):(18)$$K_{11}(s)=\frac{N_{11}(s)}{D(s)},K_{12}(s)=\frac{N_{12}(s)}{D(s)},K_{21}(s)=\frac{N_{21}(s)}{D(s)},K_{22}(s)=\frac{N_{22}(s)}{D(s)}$$

The numerical values of the numerators and denominators of the components of the robust controller *K*(*s*), namely, *N_11_*(*s*), *N_12_*(*s*), *N_21_*(*s*), *N_22_*(*s*), and *D*(*s*) are as follows [26,28]:(19)$$N_{11}(s)=2\cdot 10^{-5}\cdot s^{6}+2\cdot 10^{-5}\cdot s^{5}+4\cdot 10^{-4}\cdot s^{4}+0.0312\cdot s^{3}+0.0532\cdot s^{2}+0.0201\cdot s+0.0122$$
(20)$$N_{12}(s)=8\cdot 10^{-5}\cdot s^{6}+7\cdot 10^{-5}\cdot s^{5}+5\cdot 10^{-5}\cdot s^{4}+0.0422\cdot s^{3}+0.0976\cdot s^{2}+0.0143\cdot s+0.0211\;$$
(21)$$N_{21}(s)=5\cdot 10^{-5}\cdot s^{6}+6\cdot 10^{-5}\cdot s^{5}+7\cdot 10^{-5}\cdot s^{4}+0.0235\cdot s^{3}+0.0232\cdot s^{2}+0.0089\cdot s+0.0678\;$$
(22)$$N_{22}(s)=2\cdot 10^{-4}\cdot s^{6}+3\cdot 10^{-4}\cdot s^{5}+0.005\cdot s^{4}+0.351\cdot s^{3}+0.6232\cdot s^{2}+0.3015\cdot s+0.0242\;$$
(23)$$D(s)=3\cdot 10^{-4}\cdot s^{7}+2\cdot 10^{-4}\cdot s^{6}+9\cdot 10^{-4}\cdot s^{5}+0.0196\cdot s^{4}+1.4102\cdot s^{3}+2.5062\cdot s^{2}+1.2128\cdot s+0.0973\;$$

The following are the general rules for the selection of the weighting matrices in Relation (15) that characterize the robust control [26,28]:(24)$$W_{1S}(s)=\left[\begin{array}{cc} \frac{0.0036\cdot s+2.1522}{s} & 0\\ 0 & \frac{0.0036\cdot s+2.1522}{s} \end{array}\right]$$
(25)$$W_{1U}(s)=\left[\begin{array}{cc} 0.01 & 0\\ 0 & 0.01 \end{array}\right]\;$$
(26)$$W_{1T}(s)=\left[\begin{array}{cc} \frac{0.0033\cdot s+1}{0.0053\cdot s+0.75} & 0\\ 0 & \frac{0.0033\cdot s+1}{0.0053\cdot s+0.75} \end{array}\right]\;$$

Bode plots can be drawn to highlight the robustness properties of the synthesized controller. Thus, Figure 7a shows Bode plots for the transfer function *ω_ref_* to *z*_1_, while Figure 7b shows Bode plots for the transfer function *T_L_* to *z*_1_. From the analysis of these Bode plots, it can be seen that the robust control system of the PMSM ensures the stability of the closed-loop system, even in the case of parametric variations.

Figure 8 shows the evolution of the parameters of interest of the PMSM sensored control system using the classical FOC control structure with PI-type controllers, namely: rotor speed—*ω*; electromagnetic torque—*T_e_*; load torque—*T_L_*; stator currents—*i_a_*, *i_b_*, and *i_c_*; and *d*-*q* rotating reference frame *i_d_* and *i_q_*. The numerical simulations are performed by applying a sequence of step signals to the PMSM rotor speed reference as follows: *ω_ref_* = [1000, 1250, 1500, 900] rpm, together with a load torque *T_L_* with a value of de 0.5 Nm. It can be seen that the response time of the control system is 29 ms, and there is no overshoot.

Figure 9 shows the numerical simulations of the PMSM sensored control system, when both the classical PI-type controller (FOC-type strategy) and the robust-type controller are used. Thus, numerical simulations are performed after applying a sequence of step signals to the PMSM rotor speed reference as follows: *ω_ref_* = [1000, 1200, 1400, 1000] rpm, together with a load torque *T_L_* with a value of 0.5 Nm. For the PMSM control, better behavior of the robust controller is observed compared to the PI-based controller.

However, if there is a 50% variation in the load torque *T_L_* and a 50% variation in the combined rotor and load inertia *J* parameter, Figure 10 shows a comparison of the performance of the robust-type controller and the PI-type controller. It can be seen that the robust controller has the same performance as when the parameters have nominal values (see Figure 9), while the PI controller provides a response time more than 50% higher than when the PMSM parameters have nominal values.

### 4.2. Improvement in the Robust Control of PMSM Sensored Control System Using RL-TD3 Agent

In order to improve the performance of the robust controller according to [28], an RL-TD3 agent is used which, after the training phase, will provide correction signals over the *u_d_* and *u_q_* control signals of the robust controller, so that the control system consisting of the robust controller plus the RL-TD3 agent will provide superior performance for the PMSM control. The proposed block diagram of the implementation in Simulink of the robust controller in tandem with the RL-TD3 agent for the sensored control of a PMSM is shown in Figure 11.

Using the elements and structure from the MATLAB Reinforcement Learning toolbox, Figure 12 shows the RL-TD3 agent implementation subsystem. Note that the inputs to this subsystem, viz.: PMSM rotor speed *ω*, *i_q_* current, *i_qerror_* current error and speed error *ω_error_*, are observations for this type of algorithm, while the outputs are the correction signals for the *u_d_* and *u_q_* control signals and are considered actions. The generalized reward plays the role of a global criterion for optimizing the performance of the control system. The expression of the reward is given by the following relation [32,33]:(27)$$R_{RL-TD3}=-\left(5i_{qerror}^{2}+5\omega_{error}^{2}+0.1\sum_{j}{\left(u_{t-1}^{j}\right)}^{2}\right)$$

Note that in Relation (27), the term $u_{t-1}^{j}$ includes the actions generated in the previous step. The training performance for 200 epochs of the RL-TD3 agent is shown in Figure 13.

The following is a series of numerical simulations performed in MATLAB/Simulink for a PMSM with the nominal values given in Table 1 on the performance of the robust control system, but also with the enhancements of the RL-TD3 agent, where the variables of interest are those presented in Section 4.1. Therefore, Figure 14 shows the performance of the PMSM sensored robust control system for rotor speed reference *ω_ref_* = [1000, 1250, 1500, 900] rpm and load torque *T_L_* = 0.5 Nm, while Figure 15 shows the performance of the PMSM sensored robust control system plus RL-TD3 agent.

Figure 16 shows the results of the performance of the robust control system plus RL-TD3 agent for PMSM control, for a 100% increase in the nominal value of the load torque and a 50% increase in the combined rotor and load inertia *J* parameter. It can be seen that the control system maintains its performance, but with a corresponding increase in currents.

Figure 17 shows a comparison of the time evolution of the rotor speed for the PMSM sensored control system using the PI controller, the robust controller, and the robust controller plus RL-TD3 agent. It can be seen that, compared to the robust controller, the response time of the robust controller used in tandem with RL-TD3 agent decreases from 22.9 ms to 18.2 ms, while the PI controller provides a response time of 30.1 ms.

In addition to the response time and ripple of the PMSM rotor speed signal, the calculation of the DF of the PMSM rotor speed is also proposed as an element to compare the performance of PMSM control systems for sensored control structures.

The box-counting method is used to calculate DF, according to [34]. In principle, it starts from a square that can contain the signal, and the length of one side of this square is used as the unit of measurement. The side of the square is chosen so that it contains the signal, and the size of the side of the square is chosen as a power of 2 to speed up the calculations. By dividing the unit of measurement by 2, the algorithm is repeated until the value is below a predefined threshold. We can write that the form of each division has the expression 1/2*k*, where *k* is the current step. The division on the second axis is of the form $n_{k}/2^{2k}$ where the total number of occupied domains on the scale given by the initial signal is denoted by *n_k_*. In the current step, the values obtained for each of the two axes using the algorithm described above are retained, namely the coordinates of a point *M_k_*(*x*,*y*), represented in the Cartesian system by logarithmic coordinates. The sequence of points *M*_1_, *M*_2_, …, *M*_n_ is calculated sequentially at each step. The stop condition is when the set threshold value has been exceeded. The slope of a line closest to points *M*_1_, *M*_2_, …, *M*_n_ is calculated by least squares, and the value obtained represents the DF of the initial signal. In the MATLAB environment, using the command “[n,r] = boxcount(signal,‘slope’)”, the vectors of the two dimensions are obtained according to the algorithm described above.

Moreover, by using the commands “df = −diff(log(n))./diff(log(r))” and “[‘DF = ’num2str(mean(df(1:length(df)))) ‘+/− ’num2str(std(df(1:length(df))))])” the coordinates of the *M_k_* points in logarithmic coordinates are obtained, as well as the slope of the line closest to the points provided by the algorithm representing DF.

Figure 18, Figure 19 and Figure 20 show the DF of the rotor speed signal when using PI controller, robust controller, and robust controller plus RL-TD3 agent for PMSM sensored control system.

Table 2 shows the comparative performance of the proposed controllers for the PMSM sensored control system. The elements that are compared are response time, rotor speed ripple and DF of the rotor speed signal. The performance improvement is clearly seen when using a robust controller compared to the case of using the classical PI controllers in the sensored control structure of the PMSM. Moreover, the improvement in the performance of the PMSM control system can be further enhanced by using the robust controller in tandem with an RL-TD3 agent. Note that by analyzing the obtained values of the DF of the rotor speed signal for the three cases of controllers presented, the hypothesis presented in [34] is verified, i.e., when using a controller with superior performance, the DF of the controlled signal is high.

## 5. PMSM Sensorless Control System Using Improved ESO-Type Observer Variants

This section presents the sensorless control of the PMSM, where the controller used is the one described in Section 4.2, i.e., a robust controller working in tandem with an RL-TD3 agent. Once the controller is fixed, there is the problem of estimating the speed of the PMSM rotor by means of a speed observer. Due to the structure of the nonlinear equations of the PMSM, an ESO-type observer is chosen according to [29]. This section also proposes improvements to PMSM rotor speed estimation, where the element of comparison is the rotor speed provided by the speed sensor in the PMSM sensored control structure. Therefore, first of all, the ESO-type observer, as its name implies, extends the estimated states by an additional state, and from the description of the ESO-type observer equations, it is also possible to estimate the load torque *T_L_*. In order to achieve an improvement in the estimated speed of the PMSM rotor, it is proposed to use a multiple NN, which uses multiple NNs trained over portions of the load torque variation, and selected accordingly by a stateflow to provide an estimate of the load torque. A second way to improve PMSM rotor speed estimation is to use an RL-TD3 agent that, after training, provides signals that overlap the ESO-type observer output to provide an estimate of the PMSM rotor closer to the value provided for the PMSM sensored control structure.

### 5.1. ESO-Type Observer Description

To estimate the PMSM rotor speed using the ESO-type observer, we start from the form of the PMSM nonlinear description equations. These equations were described in Section 2, but can be written in the general form given by Equation (28).
(28)$$y^{\left(n\right)}(t)=f\left(y(t),\dot{y}(t),\ldots ,y^{\left(n-1\right)}(t),d(t),t\right)+bu(t)$$

Equation (28) describes a monovariable nonlinear system in which perturbations occur, where *y*^(*l*)^ denotes the derivative *l* of the output of the system *y*, *u* denotes the input to the system *y*, and *d* denotes the perturbation acting on the system *y*.

By denoting $x_{1}=y$, $x_{2}=\dot{y},\ldots ,x_{n}=y^{\left(n-1\right)}$, the following form is obtained:(29)$$\left\{\begin{array}{c} {\dot{x}}_{i}=x_{i+1},i=1,\ldots ,n-1\\ {\dot{x}}_{n}=f\left(x_{1},x_{2},\ldots ,x_{n},d,t\right)+bu \end{array}\right.$$

An additional state is also selected in the following form:(30)$$\left\{\begin{array}{c} x_{n+1}=f\left(x_{1},x_{2},\ldots ,x_{n},d,t\right)\\ {\dot{x}}_{n+1}=h(t) \end{array}\right.$$
which gives the form of the function *h*(*t*):(31)$$h(t)=\dot{f}\left(x_{1},x_{2},\ldots ,x_{n},d,t\right)$$

Note that in the descriptions given by the previous equations, the variable *f* includes the concentrated disturbances acting on the system.

Starting from the equations of the PMSM nonlinear system in Section 2, and adding the previous transformations, the equations of an ESO-type observer can be synthesized in the following form [29]:(32)$$\left\{\begin{array}{c} {\dot{\hat{x}}}_{i}={\hat{x}}_{i+1}+\beta_{i}(y-{\hat{x}}_{1}),i=1,\ldots ,n\\ {\dot{\hat{x}}}_{n+1}=\beta_{n+1}(y-{\hat{x}}_{1}) \end{array}\right.$$

Thus, Relation (32) describes the component equations of the system describing the ESO-type observer which, based on a minimum of information, will provide the estimate of the states of the system together with the extended state including the *f* term.

By adjusting the above relation to estimate the PMSM rotor speed, the following relation can be written:(33)$$\frac{d\omega }{dt}=\frac{1}{J}\left(\frac{3}{2}n_{p}\lambda_{0}i_{q}-T_{L}-B\omega \right)$$
where the term *f*, which includes the generalized disturbances acting on the system of equations describing the operation of the PMSM, is expressed as follows:(34)$$f=-\frac{B}{J}\omega -\frac{T_{L}}{J}$$

Substituting Relation (34) into Relation (33) gives the following equation:(35)$$\frac{d\omega }{dt}=f+bi_{q}$$

If in the previous relations *x*_1_ = *ω*, *x*_2_ = *f* and $\dot{f}=h$, the matrix expression of the system containing the PMSM rotor speed is obtained as follows:(36)$$\left\{\begin{array}{l} \dot{x}=Ax+Bu+Εh\\ y=Cx \end{array}\right.$$
where the system matrices (36) are given by:(37)$$A=\left[\begin{array}{cc} 0 & 1\\ 0 & 0 \end{array}\right];\;B=\left[\begin{array}{c} b\\ 0 \end{array}\right];\;C=\left[\begin{array}{cc} 1 & 0 \end{array}\right];\;E=\left[\begin{array}{c} 0\\ 1 \end{array}\right]$$

Based on this, the ESO-type observer structure can be written as follows:(38)$$\left\{\begin{array}{l} \dot{x}=Ax+Bu+L(x_{1}-z_{1})\\ y=Cz \end{array}\right.$$

The bandwidth of the ESO-type observer system is denoted by *ω*_0_. The poles of this system are assigned the value −*ω*_0_, and to determine the gain of the observer *L*, the identification is derived from the equality of polynomials as follows [29]:(39)$$\lambda (s)=s^{2}+L_{1}s+L_{2}={\left(s+\omega_{0}\right)}^{2}$$
(40)$$\left\{\begin{array}{c} L_{1}=2\omega_{0}\\ L_{2}=\omega_{0}^{2} \end{array}\right.\;$$
(41)$$L=\left[\begin{array}{c} 2\omega_{0}\\ \omega_{0}^{2} \end{array}\right]\;$$

For fast convergence, the chosen *ω_0_* is equal to 900, and in the implementation in Figure 21, it can be seen that the PMSM rotor speed is estimated by the state variable *x*_1_ of the observer, and the term *f* is estimated by the state variable *x*_2_ of the observer.

Furthermore, considering the estimated PMSM rotor speed, the estimated PMSM rotor angle can also be obtained as:(42)$$\hat{\theta }(t)=\theta (t_{0})+\int_{t_{0}}^{t}\hat{\omega }(t)dt$$
where: $\theta (t_{0})$ is the initial value of the rotor position relative to a fixed reference point.

The estimate of the load torque can also be obtained in the form of the following equation, where *x*_1_ and *x*_2_ represent the states of the ESO-type observer:(43)$${\hat{T}}_{L}=J\left(-\frac{B}{J}x_{1}-x_{2}\right)$$

In the following section, a method is proposed to improve the estimation of the PMSM rotor speed, by replacing Equation (43), which was used to estimate the load torque *T_L_* with a multiple NN trained on the specific portions of the load torque variation.

### 5.2. Load Torque Estimation Using Multiple NN

*T_L_* load torque estimation is performed using multiple NNs, each of which is a two-layer feed-forward network [30,31]. The Neural Net Fitting application from MATLAB is used in the implementation and proper training of each NN over 5 load torque ranges *T_L_*, i.e., between 0 Nm and 1 Nm. After the numerical simulation on a predetermined interval in the load torque range specified above, the input data of each NN, namely the PMSM rotor speed *ω*, the derivative of the PMSM rotor speed *dω*/*dt*, *i_d_* and *i_q_* currents from *d*-*q* frame, but also the output of each NN given by the load torque required to generate and train it, were recorded. Therefore, in the Neural Net Fitting application, the input data, represented by sets of 500,000 values for each variable, is divided into 70% data needed for training, 15% data needed for validation, and 15% data needed for testing.

The algorithm chosen for training, validation, and testing is Bayesian regularization. This type of algorithm minimizes a linear combination of squared errors and weights. It also modifies the linear combination so that, at the end of training, the resulting network has good generalization qualities. After generating the required 5 NNs in the selected load torque range, each NN is selected using a stateflow implemented in MATLAB [30,31]. Figure 22 shows the MATLAB/Simulink subsystem implementation of multiple NN for load torque estimation and selection. Implementation of a NN for load torque estimation in the Neural Net Fitting app from MATLAB is shown in Figure 23. Details of this implementation, in terms of the block diagram of a NN for load torque estimation, represented by two-layer feed-forward network with sigmoid hidden neurons (150) and linear output neurons for regression tasks, is shown in Figure 23a, while Figure 23b shows the Simulink NN exported model for load torque estimation, and this stage is performed using the Neural Net Fitting application after the process of training, validation and testing of each NN.

The number of epochs used in the training phase is 1000, and the learning rate is 0.1. Figure 24 shows the best validation performance, which represents the mean square error (MSE) values obtained on the entire data set selected for training, validation and testing during the iterative calculation process. Therefore, it can be seen that, at epoch 518, a MSE value of 0.00031339 is obtained from the total computation period of the Bayesian regularization algorithm. Figure 25 shows the evolution of the gradient in the iterative calculation algorithm, and it can be seen that, when applying the NN algorithm that provides the load torque estimation at epoch 524, the gradient value is 0.00001489.

Figure 26 shows the sample error distribution in the iterative calculation process. It can be seen that the sample error corresponding to an NN for load torque estimation is distributed between values of ±0.01. The regression analysis diagrams of the training sample data, test sample data, and the entire data set during the application of the Bayesian regularization type algorithm, are shown in Figure 27. It can be seen that the correlation factor R value corresponding to a NN for load torque estimation is around the value of 0.94.

Figure 28 shows the load torque and estimated load torque using multiple NN estimation time evolution after numerical simulation of the PMSM control system for the following load torque sequence [0.1, 0.7, 0.9, 0.5, 0.3] Nm. It can be seen that the multiple NN estimation is very good, which creates the conditions for obtaining superior PMSM rotor speed estimation results when the load torque is provided by multiple NNs in the ESO observer structure.

### 5.3. Improved Estimation of PMSM Rotor Speed Using ESO-Type Observer and RL-TD3 Agent

Similar to using an RL-TD3 agent to improve the performance of the PMSM control system, an agent whose implementation is shown in Figure 29 is selected and its reward is defined by the following relation [32,33]:(44)$$r_{ESO}=-\left(Q\omega_{error}^{2}+R\sum_{j}{\left(u_{t-1}^{j}\right)}^{2}\right)$$
where *Q* = 0.5, *R =* 0.1, and $u_{t-1}^{j}$ is the history of previous weighted actions.

Figure 30 shows the evolution of the performance of the RL-TD3 agent algorithm during the training phase over a period of 200 episodes. After the RL-TD3 agent training stage, it will cooperate with the ESO-type observer in the sense that the RL-TD3 agent will provide correction signals that overlap the value of the rotor speed estimation provided by the ESO-type observer itself, thus achieving superior performance to the PMSM rotor speed estimation. Thus, by implementing this observer variant, an improvement in rotor speed estimation is achieved for the sensorless variant of the PMSM control structure.

## 6. Numerical Simulation of the PMSM Sensorless Control System

MATLAB/Simulink implementation of the proposed PMSM sensorlesss control system based on robust controller combined with RL-TD3 agent for improvement in *u_d_* and *u_q_* command using improved ESO-types version observers for speed estimation is shown in Figure 31. Figure 32 shows the MATLAB/Simulink subsystem implementation for ESO-type basic version observer for PMSM rotor speed estimation.

For an evolution of the PMSM rotor speed, reference *ω_ref_* = [500, 800, 1100, 700] rpm and load torque *T_L_* = [0.1, 0.7, 0.9, 0.5, 0.3] Nm over the time interval 0–1s, sequenced as [0, 0.25, 0.5, 0.75, 1] s, Figure 33, Figure 34, Figure 35 and Figure 36 show the time evolution parameters of PMSM sensorless control system based on a robust controller with an RL-TD3 agent for four versions of the ESO-type observer for rotor speed estimation presented in Section 5.

The comparison between the time evolution of the PMSM rotor speed estimates using the four observer variants and the time evolution of the measured PMSM rotor speed is shown in Figure 37.

Table 3 summarizes the comparative performance of the four observer variants of the PMSM rotor speed and measured PMSM rotor speed as a function of the response time and estimated speed ripple. Note that the error ripple for an analyzed variable, relative to the reference variable, can be associated with the root mean square (RMS) value of the error between the two signals. While, in Table 2, the error ripple was calculated for the error signal given by the reference rotor speed and the measured rotor speed, in Table 3 the error ripple is calculated for the error signal given by the measured rotor speed and the estimated rotor speed. It can be noted that the best performance in PMSM rotor speed estimation, in terms of the lowest estimated response time and speed ripple, is obtained when using the ESO-type observer with the load torque estimated using the multiple NN observer combined with the RL-TD3 agent.

## 7. Conclusions

Naturally, PMSM operation is subject to a number of perturbations, including both changes in internal parameters (e.g., stator resistance *R_s_* and combined inertia of rotor and load *J*) and changes in load torque *T_L_*. In order to maintain the performance of the control system over a relatively wide range of variation of the above types of parameters, a PMSM control system based on a robust controller is used. The article also presents the synthesis of the robust controller, its implementation in MATLAB/Simulink, and an improved version using an RL-TD3 agent that works in tandem with the robust controller to achieve superior performance of the PMSM sensored control system.

Regarding the sensorless control of the PMSM, starting from a basic structure of an ESO-type observer, four variants of observers are presented, obtained by combining the use of a multiple NN and an RL-TD3 agent.

Future work proposes the implementation in real time and the optimization of the performance, both in terms of control and in terms of PMSM rotor speed estimation, by modifying the robust control weights accordingly, and implementing multiple NN and reward of the RL-TD3 agent.

## Figures and Tables

**Figure 1 sensors-23-05799-f001:**
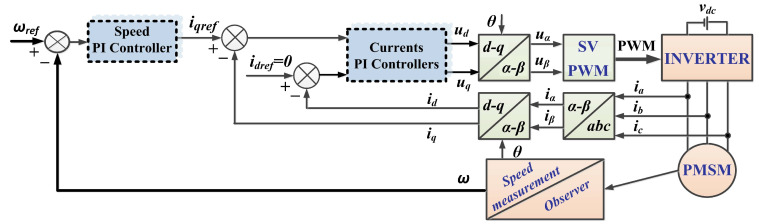
Schematic diagram of the FOC-type strategy for the PMSM control system based on PI-type controllers.

**Figure 2 sensors-23-05799-f002:**
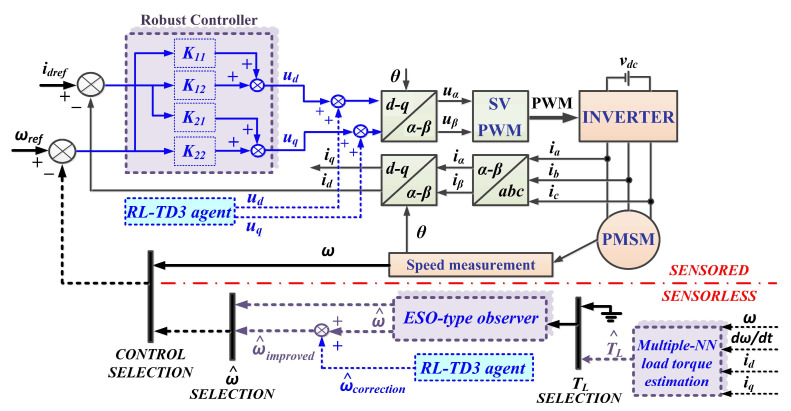
Schematic diagram of the proposed PMSM sensorless control system based on a robust controller and an RL-TD3 agent algorithm, using an improved ESO-type observer combined with a multiple NN load torque estimation and an RL-TD3 agent for speed estimation.

**Figure 3 sensors-23-05799-f003:**
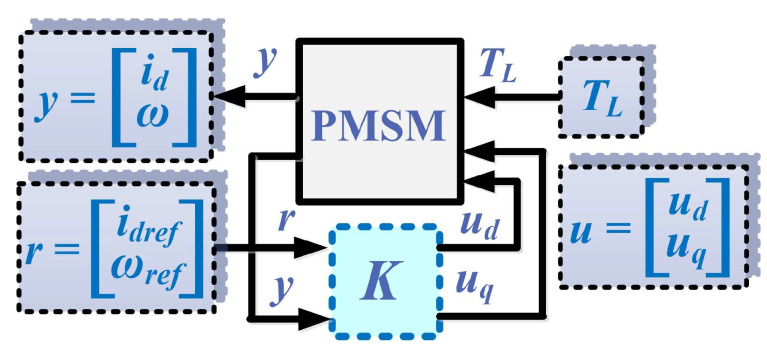
Schematic for the PMSM robust control.

**Figure 4 sensors-23-05799-f004:**
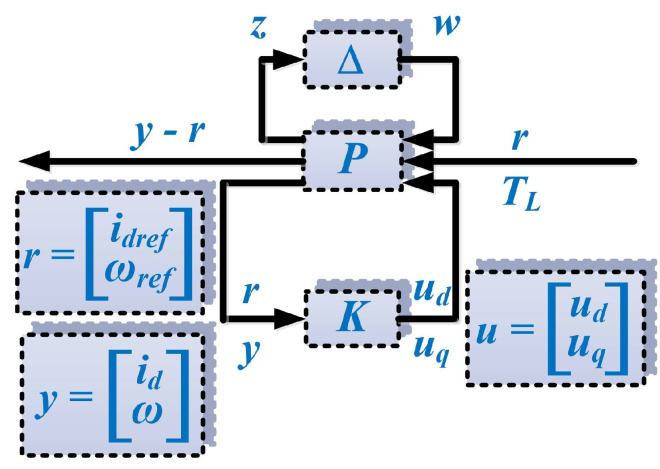
Schematic for the generalized plant.

**Figure 5 sensors-23-05799-f005:**
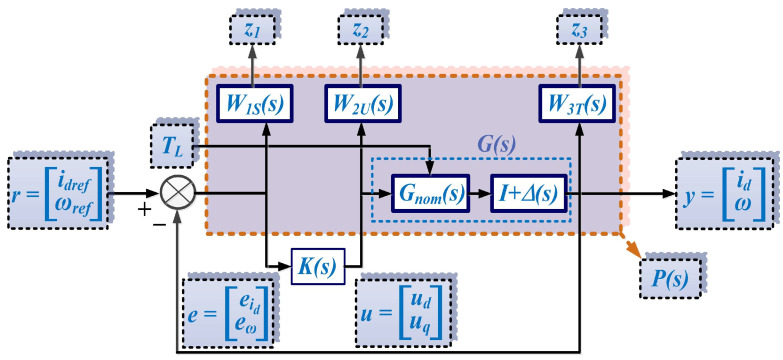
Schematic of the extended system.

**Figure 6 sensors-23-05799-f006:**
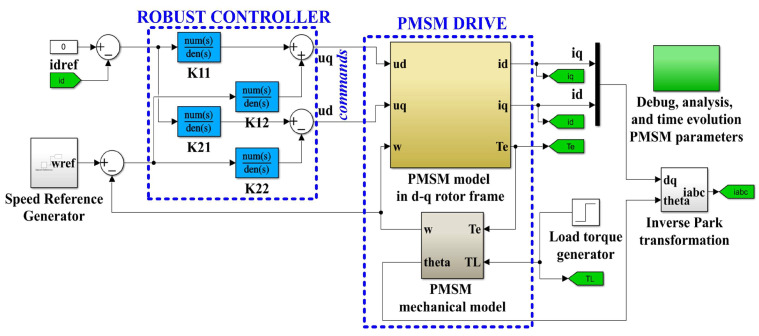
Diagram of MATLAB/Simulink model block for the proposed PMSM sensored control system using a robust controller.

**Figure 7 sensors-23-05799-f007:**
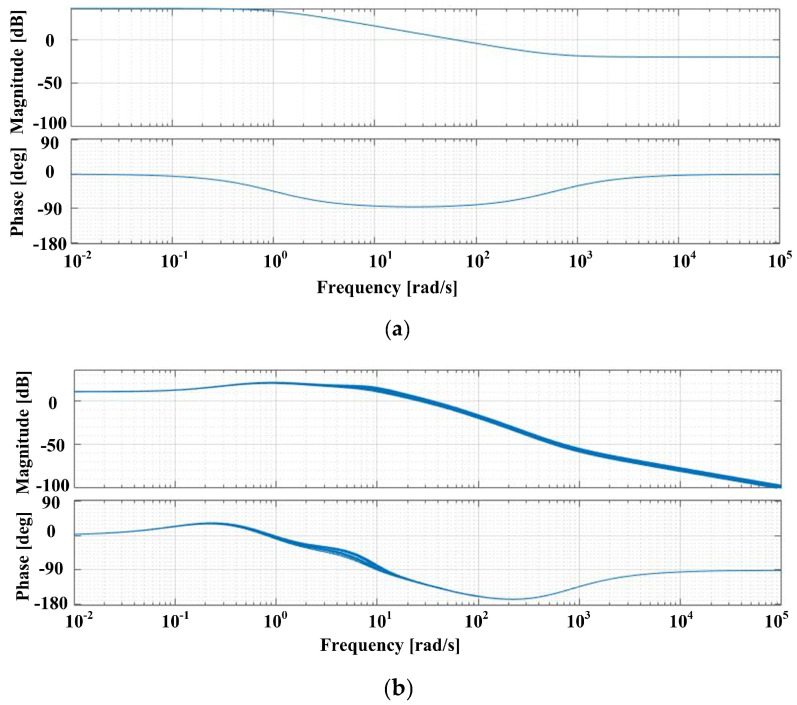
Graphical representation for Bode plots: (**a**) the transfer function from *ω_ref_* to *z*_1_; and (**b**) the transfer function from *T_L_* to *z*_1_.

**Figure 8 sensors-23-05799-f008:**
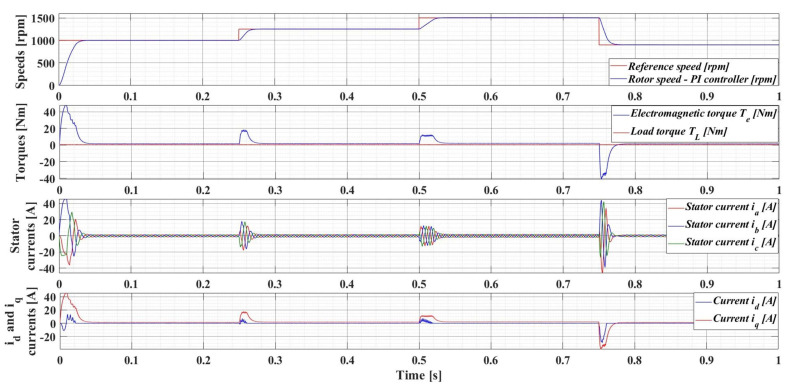
Time evolution parameters of the PMSM sensored control system using a PI controller for rotor speed reference *ω_ref_* = [1000, 1250, 1500, 900] rpm and load torque *T_L_* = 0.5 Nm.

**Figure 9 sensors-23-05799-f009:**
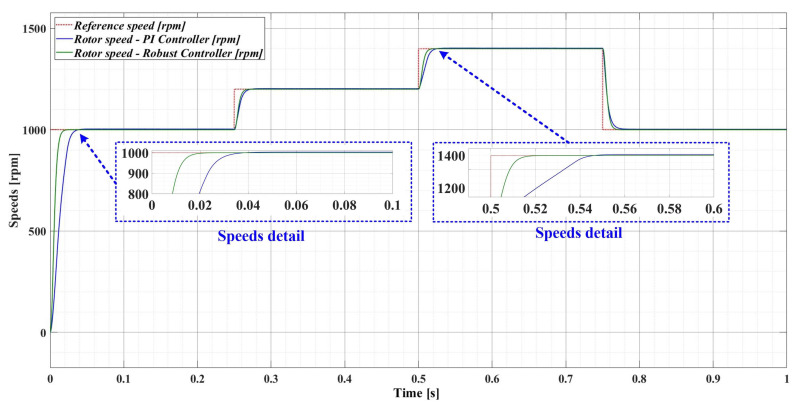
Comparison of the time evolution of the rotor speed for the PMSM sensored control system using the PI controller and the robust controller, for a rotor speed reference *ω_ref_* = [1000, 1200, 1400, 1000] rpm and load torque *T_L_* = 0.5 Nm.

**Figure 10 sensors-23-05799-f010:**
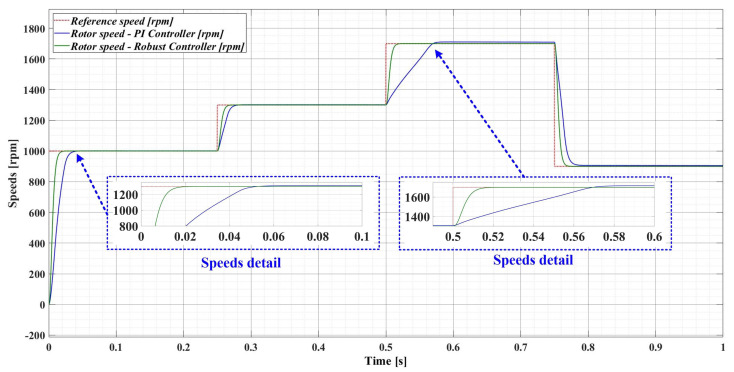
Comparison of the time evolution of the rotor speed for the PMSM sensored control system using the PI controller and the robust controller, for a rotor speed reference *ω_ref_* = [1000, 1300, 1700, 900] rpm and load torque *T_L_* = 1 Nm.

**Figure 11 sensors-23-05799-f011:**
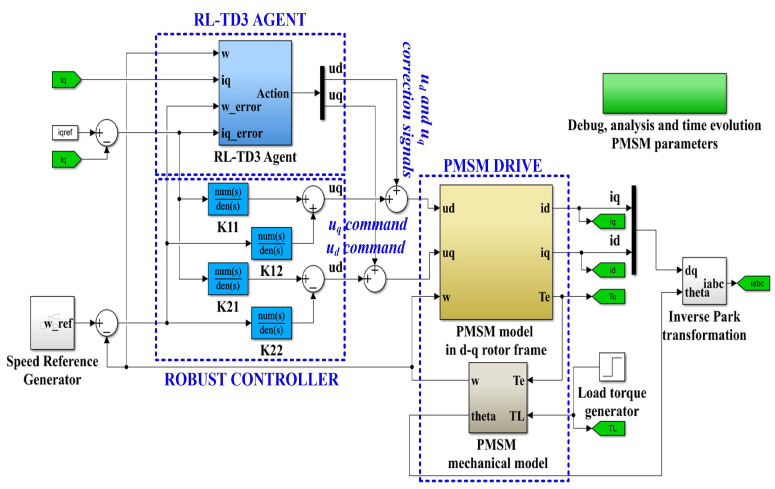
Block diagram of the MATLAB/Simulink implementation of the proposed PMSM sensored control system using a robust controller plus RL-TD3 agent.

**Figure 12 sensors-23-05799-f012:**
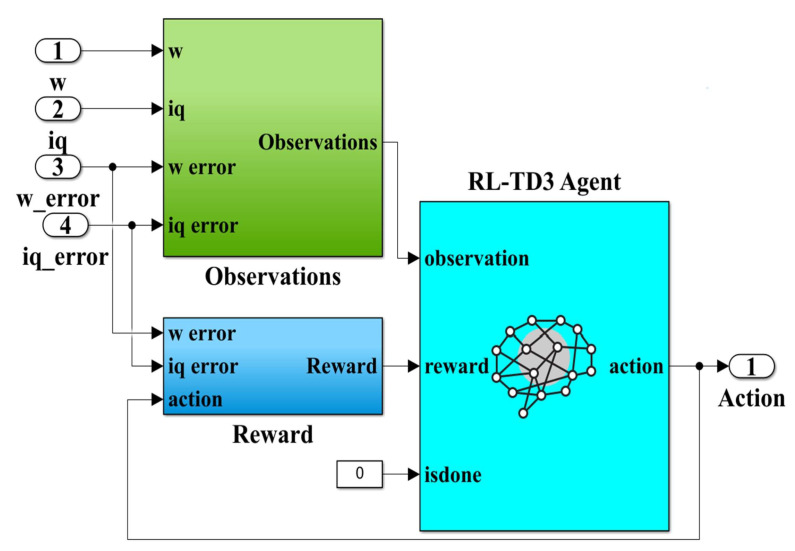
MATLAB/Simulink subsystem implementation of the RL-TD3 agent for improvement in robust controller *u_d_* and *u_q_* command.

**Figure 13 sensors-23-05799-f013:**
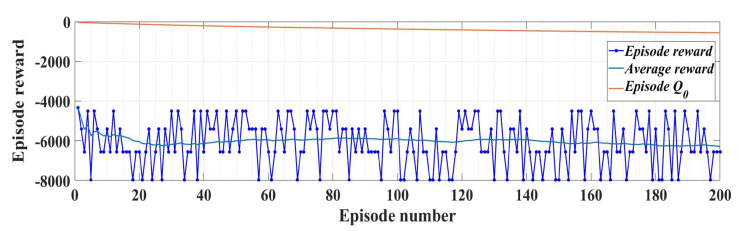
RL-TD3 agent performance evolution for *u_d_* and *u_q_* command improvement.

**Figure 14 sensors-23-05799-f014:**
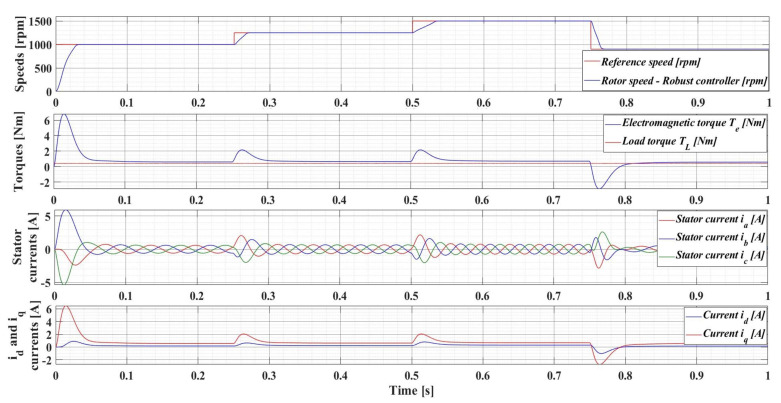
Time evolution parameters of the PMSM sensored control system using robust controller, for a rotor speed reference *ω_ref_* = [1000, 1250, 1500, 900] rpm and load torque *T_L_* = 0.5 Nm.

**Figure 15 sensors-23-05799-f015:**
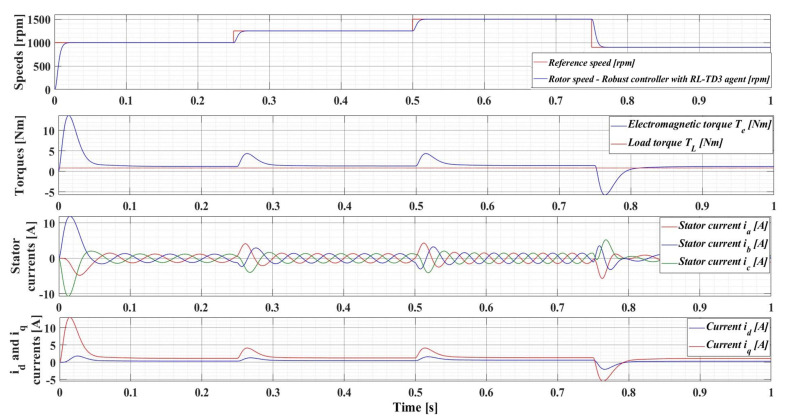
Time evolution parameters of the PMSM control system based on robust controller with RL-TD3 agent, for rotor speed reference *ω_ref_* = [1000, 1250, 1500, 900] rpm and load torque *T_L_* = 0.5 Nm.

**Figure 16 sensors-23-05799-f016:**
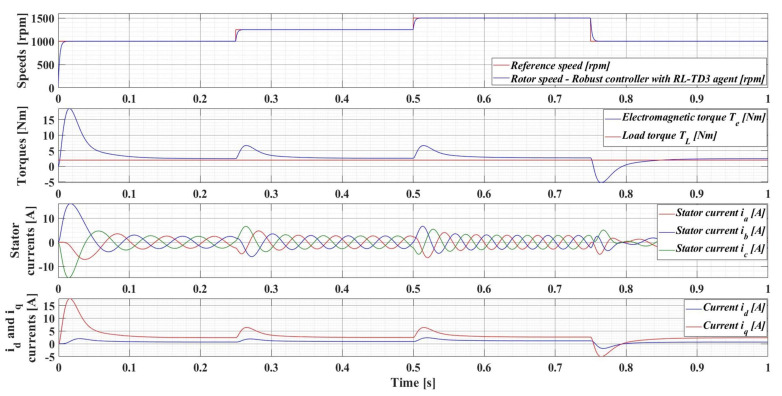
Time evolution parameters of the PMSM sensored control system using robust controller plus RL-TD3 agent, for a rotor speed reference *ω_ref_* = [1000, 1250, 1500, 900] rpm, load torque *T_L_* = 1 Nm, and 50% increase in *J* parameter.

**Figure 17 sensors-23-05799-f017:**
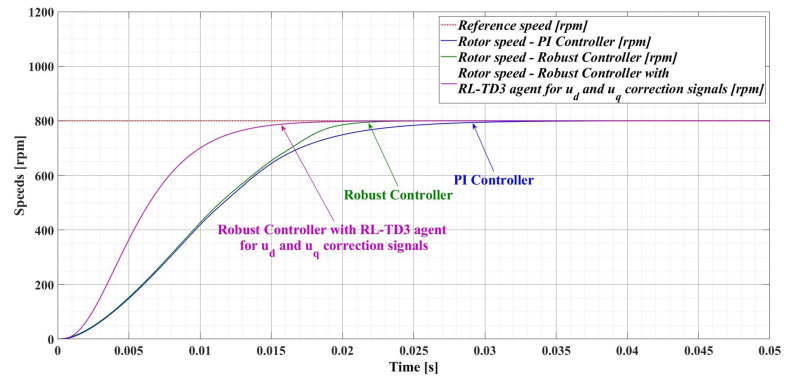
Comparison of the time evolution of rotor speed for the PMSM sensored control system using PI controller, robust controller, and robust controller plus RL-TD3 agent.

**Figure 18 sensors-23-05799-f018:**
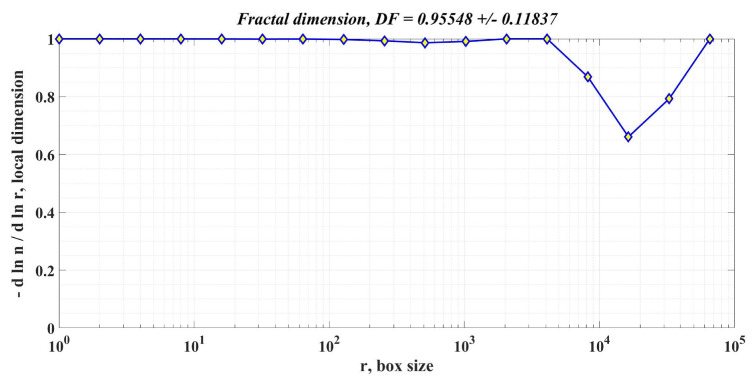
DF of the rotor speed signal in the case of using the PI controller for the PMSM control system.

**Figure 19 sensors-23-05799-f019:**
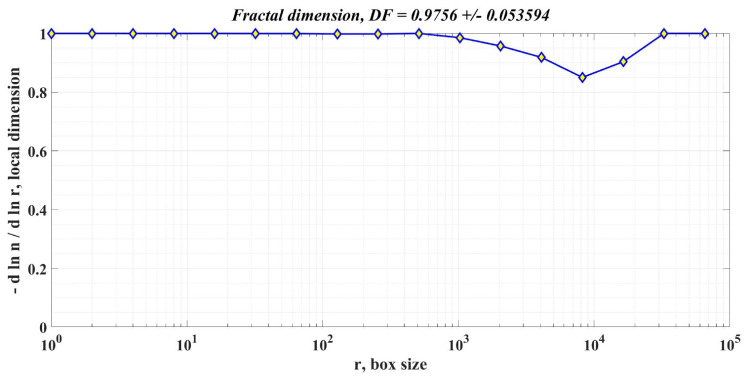
DF of the rotor speed signal in the case of using the robust controller for the PMSM control system.

**Figure 20 sensors-23-05799-f020:**
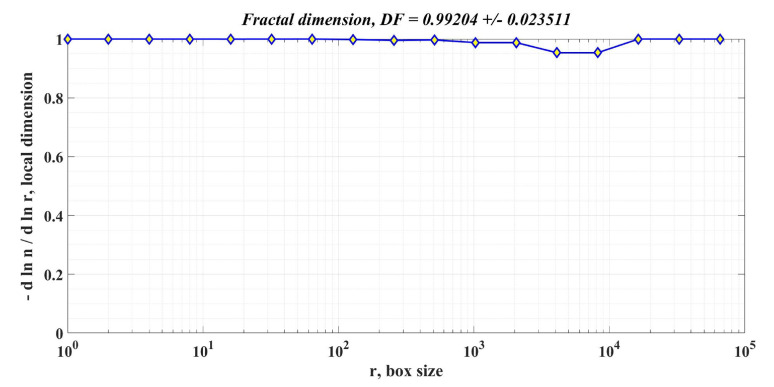
DF of the rotor speed signal in the case of using the robust controller combined with the RL-TD3 agent for the PMSM control system.

**Figure 21 sensors-23-05799-f021:**
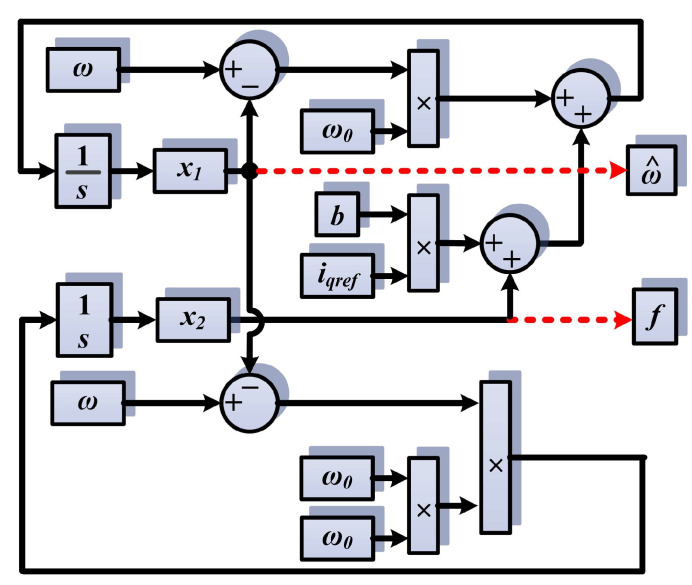
ESO-type observer block diagram. The red dotted lines highlight the output quantities of the observer.

**Figure 22 sensors-23-05799-f022:**
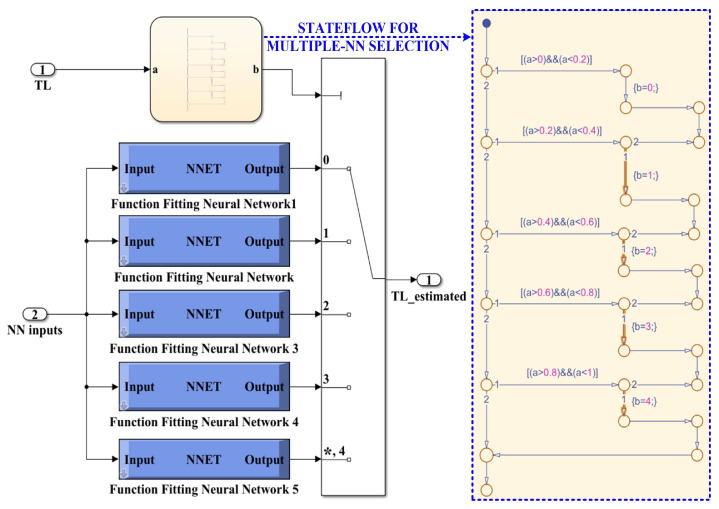
MATLAB/Simulink subsystem implementation of multiple NN for load torque estimation and selection using stateflow. “*” is specific to the multi switch block in Simulink and represents the default selection.

**Figure 23 sensors-23-05799-f023:**
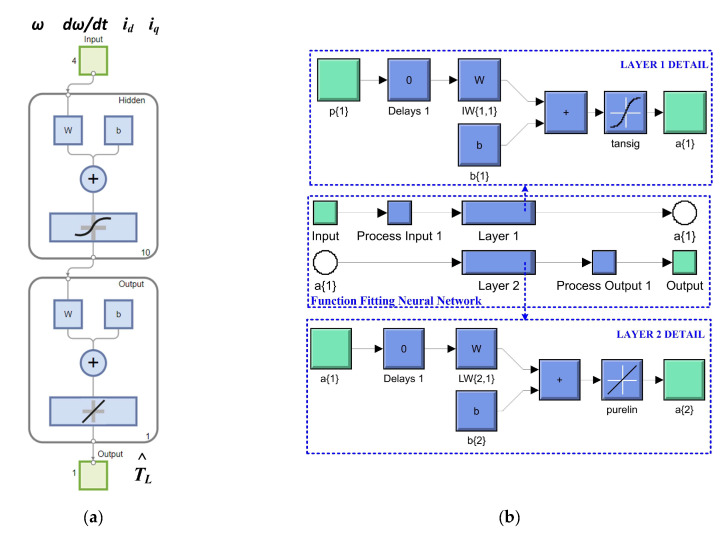
Implementation of an NN for load torque estimation in Neural Network Fitting toolbox from MATLAB: (**a**) block diagram of a NN for load torque estimation; (**b**) Simulink NN exported model for load torque estimation.

**Figure 24 sensors-23-05799-f024:**
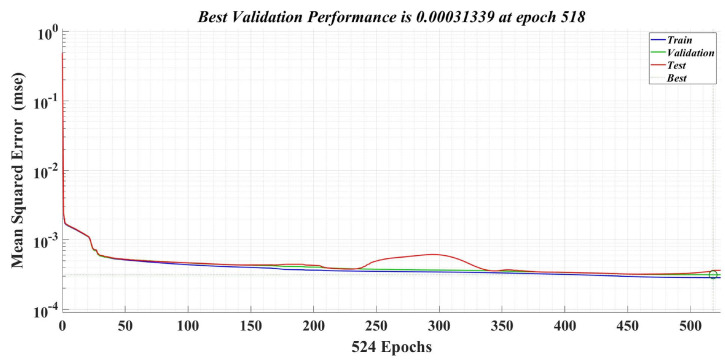
Best performance validation of a neural fitting network for load torque estimation.

**Figure 25 sensors-23-05799-f025:**
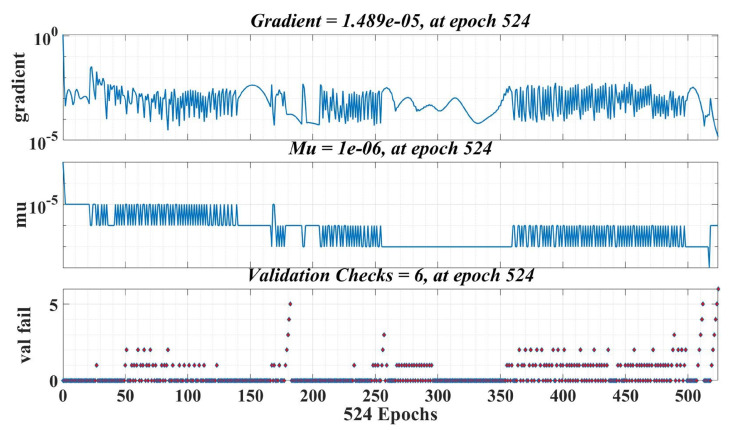
Training stage of a neural fitting network for load torque estimation.

**Figure 26 sensors-23-05799-f026:**
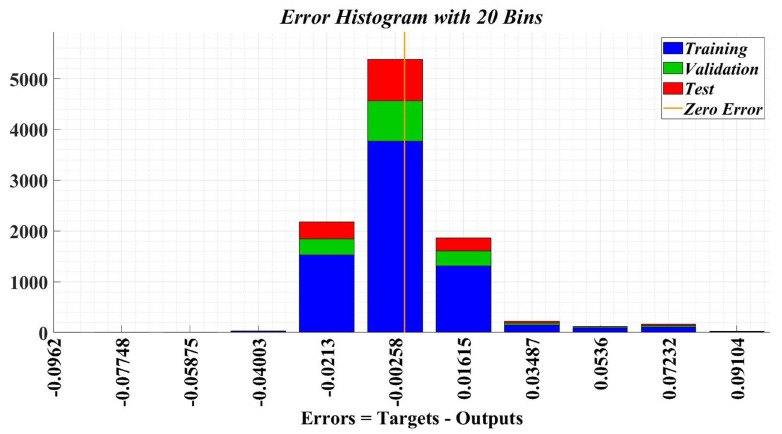
Error histogram after training stage of a neural fitting network for load torque estimation.

**Figure 27 sensors-23-05799-f027:**
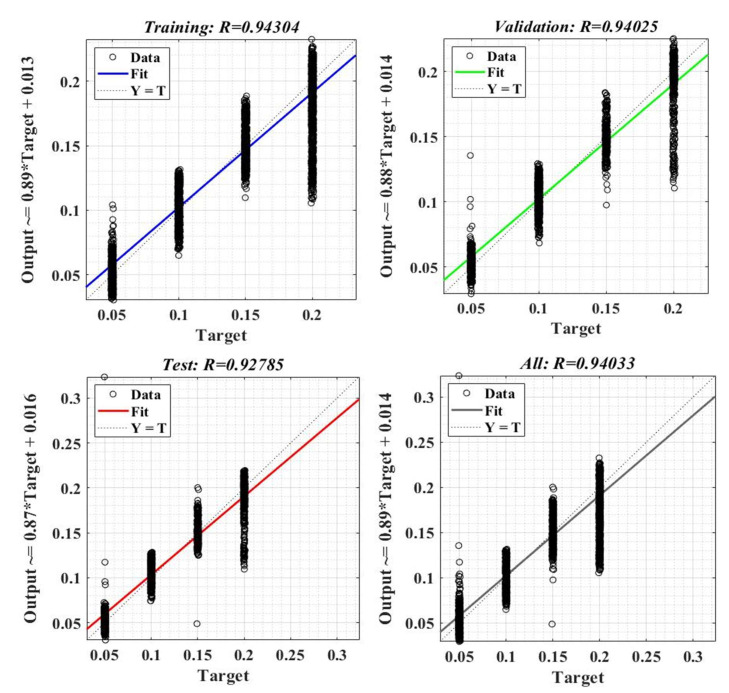
Regression factor after training, validation, and testing stages of a neural fitting network for load torque estimation.

**Figure 28 sensors-23-05799-f028:**
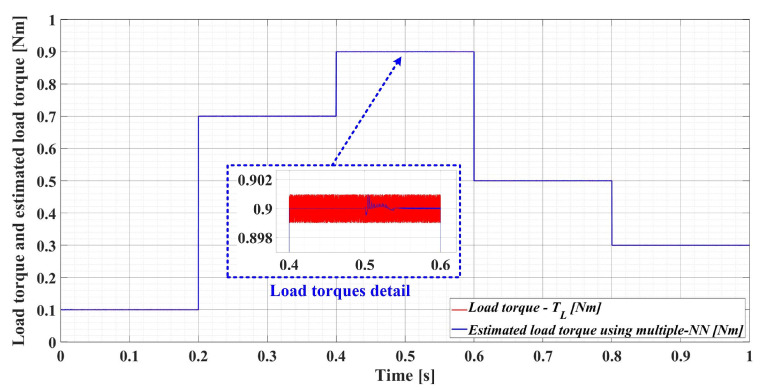
Load torque and estimated load torque using a multiple NN estimation time evolution.

**Figure 29 sensors-23-05799-f029:**
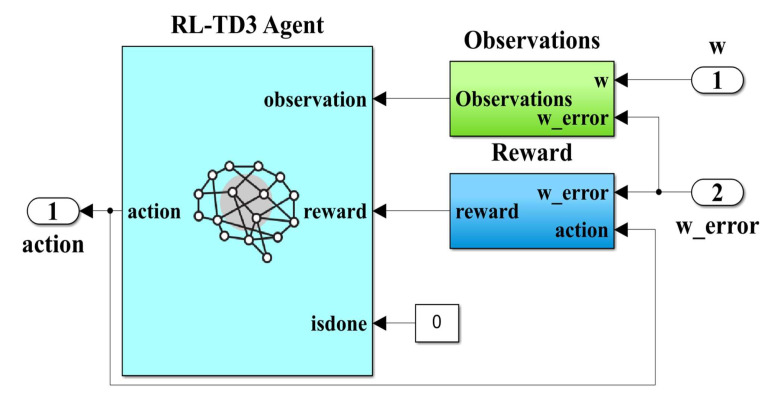
MATLAB/Simulink subsystem implementation of the RL-TD3 agent for improvement in PMSM rotor speed estimation.

**Figure 30 sensors-23-05799-f030:**
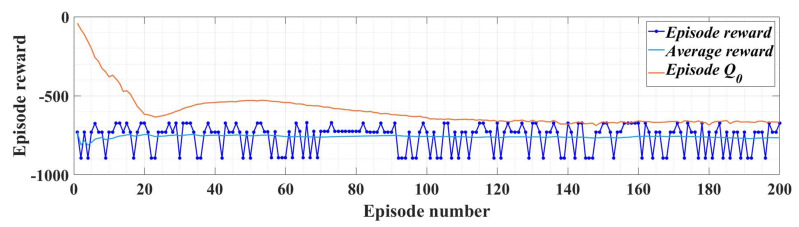
Performance evolution for the RL-TD3 agent for improvement in PMSM rotor speed estimation.

**Figure 31 sensors-23-05799-f031:**
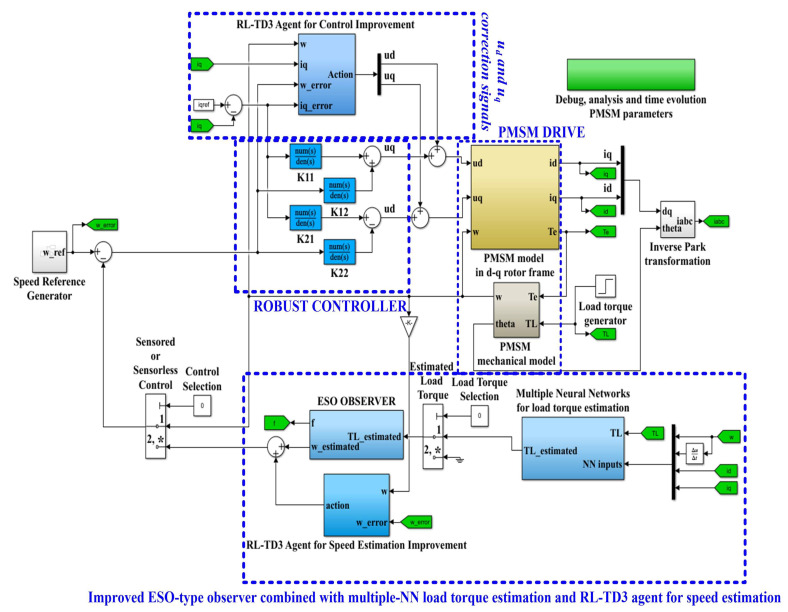
MATLAB/Simulink implementation of the proposed PMSM sensorlesss control system based on a robust controller combined with an RL-TD3 agent for improvement in *u_d_* and *u_q_* command, using improved ESO-types version observers for speed estimation. “*” is specific to the multi switch block in Simulink and represents the default selection.

**Figure 32 sensors-23-05799-f032:**
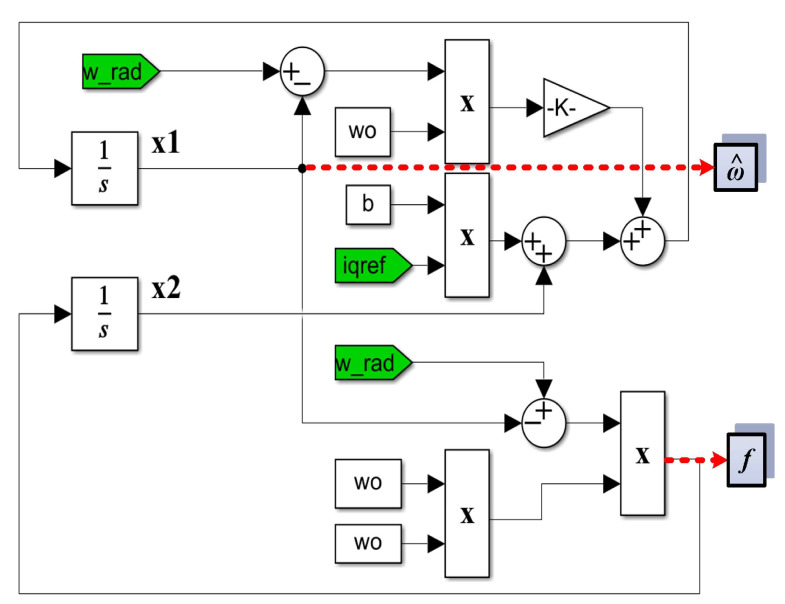
MATLAB/Simulink subsystem implementation for an ESO-type basic version observer for PMSM rotor speed estimation. The red dotted lines highlight the output quantities of the observer.

**Figure 33 sensors-23-05799-f033:**
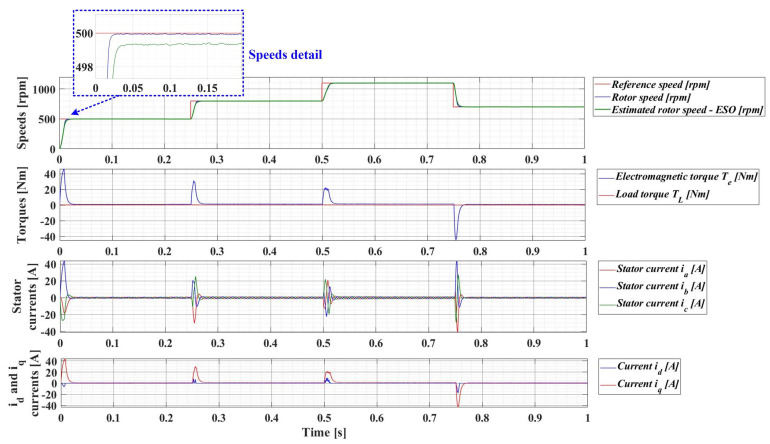
Time evolution parameters of the PMSM sensorless control system based on a robust controller with an RL-TD3 agent, using the ESO-type basic version observer for rotor speed estimation; rotor speed reference *ω_ref_* = [500, 800, 1100, 700] rpm and load torque *T_L_* = [0.1, 0.7, 0.9, 0.5, 0.3] Nm.

**Figure 34 sensors-23-05799-f034:**
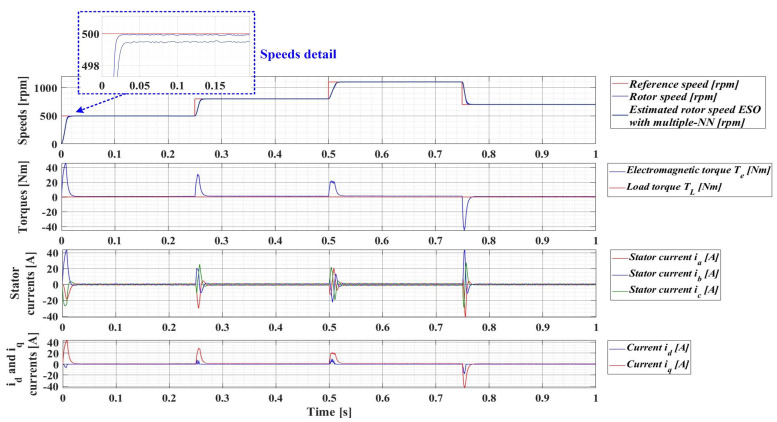
Time evolution parameters of the PMSM sensorless control system based on a robust controller with an RL-TD3 agent, using an ESO-type version observer with a multiple NN observer for rotor speed estimation; rotor speed reference *ω_ref_* = [500, 800, 1100, 700] rpm and load torque *T_L_* = [0.1, 0.7, 0.9, 0.5, 0.3] Nm.

**Figure 35 sensors-23-05799-f035:**
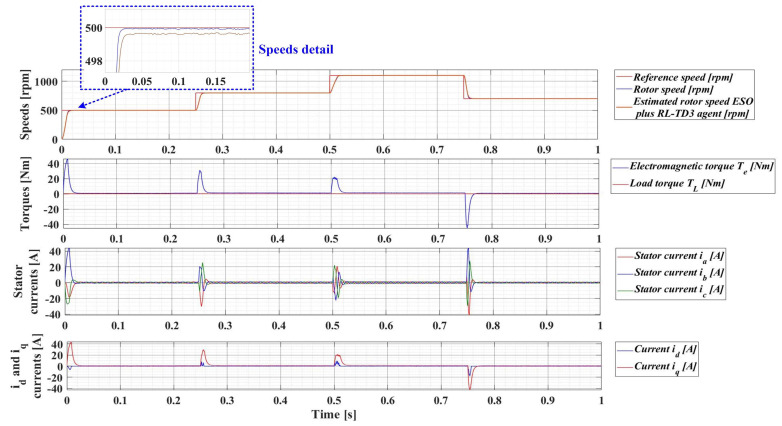
Time evolution parameters of the PMSM sensorless control system based on a robust controller with an RL-TD3 agent, using an ESO-type version observer combined with an RL-TD3 agent observer for rotor speed estimation; rotor speed reference *ω_ref_* = [500, 800, 1100, 700] rpm and load torque *T_L_* = [0.1, 0.7, 0.9, 0.5, 0.3] Nm.

**Figure 36 sensors-23-05799-f036:**
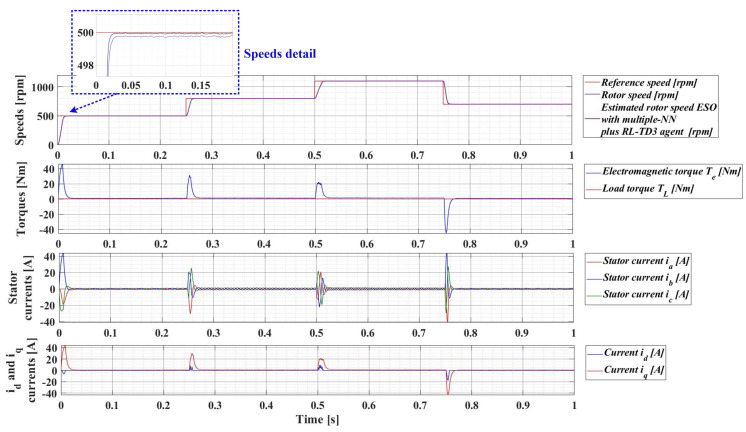
Time evolution parameters of the PMSM sensorless control system based on a robust controller with an RL-TD3 agent, using an ESO-type version with a multiple NN observer combined with an RL-TD3 agent for rotor speed estimation; rotor speed reference *ω_ref_* = [500, 800, 1100, 700] rpm and load torque *T_L_* = [0.1, 0.7, 0.9, 0.5, 0.3] Nm.

**Figure 37 sensors-23-05799-f037:**
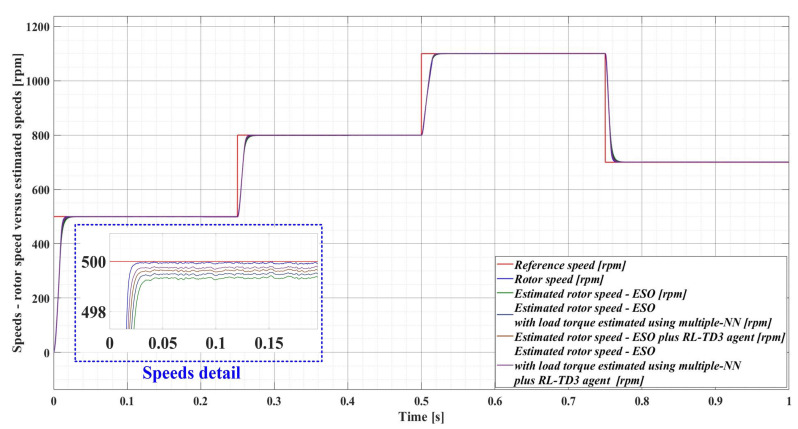
Comparison of the time evolution for the rotor speed in a sensored case, and estimated rotor speeds in sensorless cases (improved ESO-types version observers) for the PMSM control system based on a robust controller with an RL-TD3 agent.

**Table 1 sensors-23-05799-t001:** PMSM nominal parameters used in numerical simulation.

Parameter	Value	Unit
Stator resistance—*R_s_*	2.875	Ω
Inductances on *d*-*q* rotating reference frame—*L_d_* and *L_q_*	0.0085	H
Combined inertia of rotor and load—*J*	0.008	kg·m^2^
Combined viscous friction of rotor and load—*B*	0.005	N·m·s/rad
Flux induced by the permanent magnets of the rotor in the stator phases—*λ*_0_	0.175	Wb
PMSM Pole pairs number—*n_P_*	4	-

**Table 2 sensors-23-05799-t002:** Performances of proposed controllers for PMSM sensored control system.

Controller Used in PMSM Sensored Control System	Response Time [ms]	Rotor Speed Ripple [rpm]	DF of Rotor Speed Signal
PI controller	30.1	108.1	0.95548 +/− 0.118370
Robust controller	22.9	87.3	0.97560 +/− 0.053594
Robust controller combined with RL-TD3 agent	18.2	52.4	0.99204 +/− 0.023511

**Table 3 sensors-23-05799-t003:** Performances of proposed speed observers for PMSM sensorless control system.

Observer	Response Time [ms]	Improved of Response Time in Comparison with ESO-Type Basic Version Observer [%]	Estimated Speed Ripple [rpm]	Improved of Speed Ripple in Comparison with ESO-Type Basic Version Observer [%]
ESO-type basic version observer	30.7	–	3.43	–
ESO-type version observer with load torque estimated using multiple NN observer	27.4	11	2.48	27
ESO-type version observer combined RL-TD3 agent	23.5	24	1.85	46
ESO-type version observer with load torque estimated using multiple NN observer combined with RL-TD3 agent	21.1	32	1.39	59

## Data Availability

Not applicable.

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
