# Peer review of "Improved Performance for PMSM Sensorless Control Based on Robust-Type Controller, ESO-Type Observer, Multiple Neural Networks, and RL-TD3 Agent†"

_sensors, 2023, doi:10.3390/s23135799_

Round 1
Reviewer 1 Report
1. Since the topic of the article is well-researched, the cited references are not enough in this regard. So, the authors are invited to add more references to cover many related approaches over the past years. The authors should undertake a thorough literature review in introduction section to cover the state-of-the-art work, see: (a) Engineering Review, DOI: 10.30765/er.1446; (b) ISA Transactions, 2017, DOI: 10.1016/j.isatra.2017.07.003.
2.Results and discussion section must be improved by adding a discussion and comparing results with previous studies published.
3.The conclusions of the study should contain the main quantitative findings.
4.The spell-checks, grammatical and writing style errors of the paper must be improved.
5.Future studies should be provided.
The spell-checks, grammatical and writing style errors of the paper must be improved
Author Response
Dear reviewer, thanks for your recommendations.
1. The authors have published a number of papers on controllers and observers for PMSM control systems, of which the top five papers in ISI Web of Science journals are presented below, as well as over 20 papers published in IEEE Xplore conferences. The authors are reviewers and are aware of the limited number of self-citations. Therefore, although the authors have papers presenting fuzzy and backstepping control, your suggestion to include the above two papers is welcome. These have been added to the references at positions [17] and [19].
- Marcel NICOLA, Claudiu-Ionel NICOLA, Improvement of Linear and Nonlinear Control for PMSM Using Computational Intelligence and Reinforcement Learning, MDPI - Mathematics - Special Issue “Modeling and Simulation of Control System”, ISSN 1424-8220, vol. 10, issue 24, 4667, December 2022, pp. 1-34, DOI: 10.3390/math10244667; WOS:000904485300001 [IF 2.592]; [Q1]
- Marcel NICOLA, Claudiu-Ionel NICOLA, Comparative Performance Analysis of the DC-AC Converter Control System Based on Linear Robust or Nonlinear PCH Controllers and Reinforcement Learning Agent, MDPI - Sensors - Special Issue “Intelligent Control and Testing Systems and Applications”, ISSN 1424-8220, vol. 22, issue 23, 9535, December 2022, pp. 1-32, DOI: 3390/s22239535; WOS:000897394300001 [IF 3.847]; [Q2]
- Marcel NICOLA, Claudiu-Ionel NICOLA, Dan SELIȘTEANU, Improvement of PMSM Sensorless Control Based on Synergetic and Sliding Mode Controllers Using Reinforcement Learning Deep Deterministic Policy Gradient Agent, MDPI - Energies - Special Issue “Recent Advances in Smart Power Electronics”, ISSN 1996-1073, vol. 15, issue 6, 2208, March 2022, pp. 1-32, DOI: 10.3390/en15062208; WOS:000775586800001 [IF 3.252]; [Q3]
- Claudiu-Ionel NICOLA, Marcel NICOLA, Dan SELIȘTEANU, Sensorless Control of PMSM Based on Backstepping-PSO-type Controller and ESO-type Observer Using Real-Time Hardware, MDPI - Electronics - Special Issue “Hardware in the Loop, Real-Time Simulation and Digital Control of Power Electronics and Drives”, ISSN 2079-9292, vol. 10, issue 17, 2080, August 2021, pp. 1-36, DOI: 10.3390/electronics10172080; WOS:000694183000001 [IF 2.69]; [Q3]
- Marcel NICOLA, Claudiu-Ionel NICOLA, Sensorless Fractional Order Control of PMSM Based on Synergetic and Sliding Mode Controllers, MDPI - Electronics - Special Issue “Advanced Control Systems for Electric Drives”, ISSN 2079-9292, vol. 9, issue 9, 1494, September 2020, pp. 1-44, DOI: 10.3390/electronics9091494; WOS:000580287200001 [IF 2.69]; [Q2]; republicat în Cartea “Advanced Control Systems for Electric Drives” ISBN 978-3-03943-699-6 (Hbk); ISBN 978-3-03943-700-9 (PDF); https://www.mdpi.com/books/pdfview/book/3189
2. In our opinion, there are sufficient references for the problem addressed, and the comparison with most types of controllers becomes cumbersome and prohibitive. The authors prefer to point out that each controller has its own peculiarities, and the peculiarity of the controller used for the PMSM control system presented in this paper, namely a robust type controller, is that for a fixed controller structure, it achieves the rejection of disturbances and maintains the control performance for a relatively large variation of both the load torque and some intrinsic parameters of the PMSM. These elements are highlighted progressively from the controller summary to the results obtained.
3. In Table 3, we have added the improvements, expressed in percentages compared to the basic version of the ESO type observer, brought by the other proposed control variants, both in terms of time response and speed ripple.
4. We have made improvements to the text.
5. Future work proposes the implementation in real time and the optimization of the performance, both in terms of control and in terms of PMSM rotor speed estimation, by modifying the robust control weights accordingly, and implementing multiple NN and reward of the RL-TD3 agent.

Reviewer 2 Report
I tried to carefully study the submitted manuscript and I believe that it can be published in Sensors, but I have a few questions and comments
137, 145 Please give explanations for the figures 1 and 2 (α,β,abc,θ SW, PWM, K)
What is the reason for the appearance of the perturbation θ?
173 How is the load torque in the system measured? Or is this value simulated?
181 And what is included in the "structure of the regulator K"?
194 Where is transfer function of the PMSM?
234 How are the transfer functions of the controller components K obtained?
271 The fig 10 shows that the PI controller has an overshoot?
366 It is better to combine figures 18-20 into one for comparison. Why do major changes occur at box sizes larger than 10^3?
401 y(n) equation or y(l). What is bu(t)?
417 explain new symbols in equation (33)
563 Figure 37 compares the time evolution of the rotor speed using different controllers. From the graphs presented, it can be seen that the deviation in the estimated rotor speed for all cases varies within 1 rpm. And in table 3, the spread is about 3 rpm.
Can you give the quality of estimation in percentage terms (compared to the PI regulator)?
Author Response
Dear reviewer, thanks for your recommendations.
- α-β, d-q and abc are the classical notations for the direct and inverse Clake and Park transforms commonly used in diagrams describing the control of electrical drives. SV-PWM - Space Vector Pulse Width Modulation is also a common notation used in diagrams describing the control of electrical drives. The electrical angle of the PMSM is denoted by θe, and in the control diagrams of the electrical drives, θ (mechanical angle of the PMSM rotor) is usually not shown as a disturbance, but as an input quantity in the transformation blocks α-β→d-q and d-q→α-β. K is the robust controller and is a notation of the theory of robust systems, and in Sections 3 and 4 we present details on the synthesis of the controller K. References are provided [1,2,28].
- The load torque is estimated using a multiple neural network (multiple NN), and the implementation is presented in Section 5.2.
- In equations (5) and (6), the description of the PMSM is given as a differential equation together with the classical matrices A, B, C, and D, from which the transfer function can be obtained. For the robust controller synthesis, the Robust Control toolbox from the Matlab environment given in reference [25] is used, and the transfer function of PMSM is not explicitly required. However, in the paper by:
Nicola, C. -I. Nicola and M. Duţă, "Delay Compensation in the PMSM Control by using a Smith Predictor," 8th International Conference on Modern Power Systems (MPS), Cluj-Napoca, Cluj, Romania, 2019, pp. 1-6, doi:10.1109/MPS.2019.8759752,
the transfer function of the PMSM is presented both in a simplified 2nd order form and in a more complex 3rd order form.
- The robust controller K is synthesized in Sections 3 and 4 following the elements of robust systems theory and using the Matlab Robust Control toolbox for numerical synthesis.
- The purpose of Figures 9 and 10 is to show that the PI controller performs well, but over the full range of speed and torque variation, as does the robust controller. In fact, in Figure 10, PI controller for a high PMSM speed reference, the response time increases by 50% and a slight overshoot occurs. In comparison, the robust controller provides good performance over a wide range of PMSM reference speed and load torque variations.
- The box-counting method is used to calculate DF according to [34]. In principle, it starts from a square that can contain the signal, and the length of one side of this square is used as the unit of measurement. The side of the square is chosen so that it contains the signal, and the size of the side of the square is chosen as a power of 2 to speed up the calculations. By dividing the unit of measurement by 2, the algorithm is repeated until the value is below a predefined threshold. We can write that the form of each division has the expression 1/2k, where k is the current step. The division on the second axis is of the form where the total number of occupied domains on the scale given by the initial signal is denoted by nk. In the current step, the values obtained for each of the two axes using the algorithm described above are retained, namely: the coordinates of a point Mk(x,y), represented in the Cartesian system by logarithmic coordinates. The sequence of points M1, M2, ...,Mn is calculated sequentially at each step. The stop condition is when the set threshold value has been exceeded. The slope of a line closest to points M1, M2, ...,Mn is calculated by least squares and the value obtained represents the DF of the initial signal. In the Matlab environment, using the command “[n,r] = boxcount(signal,'slope')”, the vectors of the two dimensions are obtained according to the algorithm described above. Moreover, by using the commands “df = -diff(log(n))./diff(log(r))” and “['DF = 'num2str(mean(df(1:length(df)))) '+/- 'num2str(std(df(1:length(df))))])” the coordinates of the points Mk are obtained as logarithmic coordinates, and the slope of the line closest to the points provided by the algorithm, representing DF is also obtained.
From the definition of the fractal dimension above it can be seen that the slope of a line based on the method of least squares is important, and so there may be a number of points further away from the single line, in this case they occur for box sizes larger than 10^3, but what is important are the numerical results of the DF. Therefore, combining them into a single figure does not benefit the reader. It only makes it more difficult to reorganize the manuscript.
- Equation (28) represents the description of the system by a differential equation of order n. An intermediate derivative of order l este y(l). In general systems theory, this equation is transformed into a system of n differential equations of order 1 written in normal Cauchy form, with the derivatives explained "on the right". This leads to the classical form x_derivate = A*x+B*u where A is a matrix, the vector B has a single non-zero component (for a monovariable system), u is the input, b is a weighting factor (component of the vector B, usually b=1), x is the state of the system. In this way, we obtain the following equations (32). In equation (32), βi represents the design parameters of the ESO observer that end in a particular way, starting from the fulfillment of the stability condition.
- In Figure 37, the deviation is actually 1 rpmThe quality index used is the speed ripple, which is classically calculated according to the relationship , where N is the number of samples, ω is the rotor speed, and ωref is the reference rotor speed. The result is about 3rpm, because 1rpm is only in the steady state, the rest of the quantity is added by the differences (ω(i) - ωref(i)) in the transient regime.
In Table 3, we have added the improvements, expressed in percentages compared to the basic version of the ESO type observer, brought by the other proposed control variants, both in terms of time response and speed ripple.
Please see the pdf file because some equations doesn't appear here.

Round 2
Reviewer 1 Report
The paper can be accepted.